# Associations between Physical Activity, Mental Health, and Suicidal Behavior in Korean Adolescents: Based on Data from 18th Korea Youth Risk Behavior Web-Based Survey (2022)

**DOI:** 10.3390/bs14030160

**Published:** 2024-02-22

**Authors:** Suyeon Roh, Woolim Mun, Geunkook Kim

**Affiliations:** 1Department of Exercise Rehabilitation, Gachon University, Incheon 21936, Republic of Korea; dr.rohpilates@gachon.ac.kr; 2Department of Sports Rehabilitation, Jaeneung University, Incheon 22573, Republic of Korea

**Keywords:** adolescents, physical activity, mental health, suicidal behavior

## Abstract

This study aims to identify the current status and relationship between physical activity (PA), mental health, and suicidal behavior among Korean adolescents and recommend appropriate PA types and levels to lower the risk of mental health problems and suicidal behavior among adolescents. This study used the frequency of participation, three mental health factors, and four suicidal behavior factors, according to the seven types of PA, Vigorous PA (VPA), Strength training, Walks, PA on the move, and Physical education questions that 51,636 Korean adolescents responded to in the data of the 18th Korean Youth Health Behavior Survey in 2022. The results showed that physical activity levels and the mental health of female adolescents were the lowest, and the experience rate of suicidal behavior was the highest. Physical activity level and mental health were negative in the upper grades, and the experience rate of suicidal behavior was higher in the lower grades. Thus, the study proposes the following: To lower the overall risk of mental health and suicidal behavior experienced by adolescents, it is effective to encourage them to participate in physical activities which have higher exercise intensities than the Low-level Physical Activity (LPA) type of Walks and PA on the move.

## 1. Introduction

Adolescence is a time of rapid emotional and environmental change, and since it is a time when many tasks are given, and health and social help are needed, if proper adaptation and development are not achieved, it results in negative variables [1,2]. Typical negative outcomes can affect mental health, owing to psychosocial problems [3]. There are various relationships between family and school, the social environment where youths spend most of their time. The relationship between the severity of adolescent stress and loneliness is an important index for improving mental health worldwide [4,5]. Self-rated health is a representative factor that can reflect overall health in adolescence because it has a psychosocial relationship with one’s subjective mental health status and the general health of the body [2]. Therefore, self-rated health, such as stress and loneliness, can be used to identify the mental health levels of adolescents. If adolescents perceive mental health positively, they can perform various developmental tasks healthily and successfully and will become a driving force for biopsychological well-being in the adult stage [2].

However, if they do not cope with mental problems properly, they will feel unhappy and experience negative satisfaction with their life and mental health problems [6]. Adolescent mental health problems are related to suicidal ideation, planning, suicidal attempts [7,8], and hopelessness [9]. Adolescents who feel hopeless can experience suicidal ideation and suicide attempts [10,11]. Adolescents’ mental health, hopelessness, and suicidal behavior are closely related, and there is a need to carefully deal with all factors to play the role of a healthy student.

Unfortunately, Korean adolescents have experienced a drastic change in sociocultural trends. A typical sociocultural factor is excessive academic stress. In South Korea, academic expectations are much higher than in other countries. This stems from Confucianism, which emphasizes hard work and success in school and in life, and from parents’ expectations of socioeconomic success through educational success as the country has experienced multiple economic crises (i.e., the hardships of the Japanese occupation, Korean War) [12]. Theses Confucian parental expectations increase the academic stress of Korean adolescents and negatively impact their relationships with teachers and friends, affecting their mental health, including loneliness [13,14]. Currently, the economic level of the Korea has increased, but it seems to be related to the mental health of adolescents depending on the polarization of income and the residential environment (i.e., areas with high crime rates) according to the development level of each region [15].

Because of the sociocultural characteristics of Korea, it is a confusing and inadequate situation for adolescents to cope with appropriately and to form their own values and subjective self, which can have a negative impact on their mental health [16,17]. These mental health problems of Korean adolescents affect even suicidal behavior. According to the Korea National Statistical Office (KNSO), in 2021, the death rate of youths in Korea (7.1/100,000) was a serious and important subject for deliberate self-harm (suicide) [18]. Therefore, it is necessary to find and recommend ways to positively affect mental health and suicidal behaviors from early to late adolescence.

Among the methods that have been proven to be effective in solving the problems of mental health and suicidal behavior, physical activity, which means increased heart rate and breathing, is of interest because it has significantly few side effects and can entail mental advantages compared to medication [19,20,21,22]. Physical activity is also a recommended method worldwide because it has the advantage of low cost and high accessibility and affects the secretion of hormones associated with mental health such as oxytocin [23,24]. The World Health Organization (WHO) 2020 guidelines recommend moderate-to-vigorous physical activity (MVPA) for an average of >60 min per day to improve mental and cognitive health among adolescents [25]. In fact, a recent meta-analysis reported that adolescents’ participation in various types of physical activity, including aerobic exercise, yoga, and resistance training, had a positive impact on mental health [26]. In addition. studies have reported that lower levels of physical activity among adolescents are associated with an increased risk of suicidal behavior [25,27,28]. Previous research has confirmed that physical activity in adolescence has a positive impact on mental health and suicidal behavior. However, the types and levels of physical activity vary widely, and direct comparisons across physical activity factors are lacking.

On the other hand, adolescents in Korea spend a lot of time studying due to excessive competition to enter advanced school, which makes it more difficult for them to participate in physical activity [29]. Therefore, it is necessary to provide direct suggestions on effective types and levels of physical activity that can positively affect mental health and suicidal behavior.

To propose a way to solve this problem, this study aims to identify the current status of physical activity types (>60 min PA, Moderate PA, Vigorous PA, Strength training, Walks, Physical education class PA, PA on the move), mental health (self-rated health, perceived stress, and loneliness) and suicidal behavior (suicidal ideation, suicidal plan, suicidal attempt, and hopelessness) among Korean adolescents. Furthermore, by identifying the difference between the level of mental health and suicidal behavior according to the factors of physical activity, this study aims to present the standard of effective physical activity that can have a positive effect on the mental health promotion and suicidal behavior of Korean adolescents.

## 2. Materials and Methods

### 2.1. Sample

This study used data published through a cross-sectional survey, the 18th Korea Youth Risk Behavior Web-Based Survey (KYRBWS), conducted by the Korea Centers for Disease Control and Prevention (KCDC) in 2022 (approval number 11758). The sampling process at KYRBWS consisted of three stage: population stratification, sample distribution, and sample extraction. The population stratification was divided by regional districts and school levels. The sample distribution was allocated by applying a proportional distribution method to match the sample size to the population size for each stratification variable. Sample extraction was performed using stratified cluster sampling, with schools as the primary unit of sampling and classes as the secondary unit of sampling. The KYRBWS was reviewed by the Institutional Review Board (IRB) on the KCDC (2014-06EXP-02-P-A), and from 2015, the survey was conducted without review based on the Implementation Rules of the Act on Bioethics and Safety. This study was a secondary analysis of data approved by the IRB.

The subjects of this study were adolescents (*n* = 51,636) attending middle school (7th–9th grade) and high school (10th–12th grade) in South Korea. The final analysis included (i) gender: male (50.9%), female (49.1%); (ii) school type: mixed (68.5%), boys’ (15.7%), girls’ (15.7%); (iii) grade: 7th (17.8%), 8th (18.0%), 9th (18.2%), 10th (16.3%), 11th (15.4%), 12th (14.2%); (iv) location: county (7.4%), metropolis (42.8%), small/medium city (49.8%); (v) Region: capital (42.0%), Kangwon (3.7%), Chungcheong (13.5%), Honam (12.4%), Youngnam (25.9%), Jeju (2.5%).

### 2.2. Measures

The data collected in this study were anonymized using a self-reported online survey. Specifically, the survey was conducted on the following items after the adolescents visited the survey site in the classroom where wireless Internet was available during school hours, gave the survey participation number, and confirmed their participation in the survey. All survey items were reviewed by experts (7 on physical activity and 6 on mental health and, suicidal behavior) from the 9th National Health Survey Advisory Committee.

#### 2.2.1. Physical Activity

There were 7 questions about physical activity, and they are as follows:

(i) >60 min PA: “In the last 7 days, how many days has the heart rate increased more than usual or has the total of out-of-breath physical activity totaled more than 60 min per day?”; (ii) Moderate PA: “In the last 7 days, how many days have you had a moderate-intensity physical activity that is slightly out-of-breath than usual?”; (iii) Vigorous PA: “In the last 7 days, how many days have you had a high-intensity physical activity that is very out-of-breath or sweaty?”; (iv) Strength training: “In the last 7 days, how many days have you done Strength training such as push-ups, sit-ups, weight lifting, dumbbells, bars, and parallel bars?”; (v) Walks: “In the last 7 days, how many days have you walked for at least 10 min at a time?”; and (vi) PA on the move: “In the last 7 days, how many days have you walked or cycled to and from school or academy?” For each item, the number of physical activities per week was to be answered ‘0-7 times’. However, (vii) PE class PA: “How many times have you exercised directly in the playground or gym during the last 7 days?” was it to be answered ‘0, 1, 2, or more than 3 times a week’.

#### 2.2.2. Mental Health

Questions on mental health consisted of three items (self-rated health, perceived stress, and loneliness). The questions and responses are as follows: 

(i) Possible responses to the self-rated health question, “How do you feel about your usual health?” were ‘1 = very healthy’, ‘2 = healthy’, ‘3 = normal’, ‘4 = unhealthy’, and ‘5 = very unhealthy’. (ii) Possible responses to the perceived- stress question, “How much do you feel usual stress?”, were ‘1 = feel very much’, ‘2 = feel a lot’, ‘3 = feel a little’, ‘4 = do not feel much’, and ‘5 = do not feel at all’. (iii) Possible answers to the loneliness question “How often did you lonely in the last months?” were ‘1 = no loneliness at all’, ‘2 = almost no loneliness’, ‘3 = sometimes loneliness’, ‘4 = often loneliness’, and ‘5 = always loneliness’.

#### 2.2.3. Suicidal Behavior

There were four questions about suicidal behavior consisting of 4 aspects (suicidal ideation, suicidal plan, suicide attempt, and hopelessness). 

(i) Suicidal ideation: “In the last 12 months, have you seriously considered suicide?” (ii) Suicidal plan: “In the last 12 months, have you made any specific plans for suicide?” (iii) Suicidal attempt: “In the last 12 months, have you attempted suicide?” (iv) Hopelessness: “In the last 12 months, how often have you felt lonely?” Four question items were answered with ‘Yes = in the last 12 months’ and ‘No = no in the last 12 months’.

### 2.3. Statistical Analysis

SPSS 27.0 version program (IBM Corp, Armonk, NY, USA) was used for statistical analysis through the complex sampling-design method considering strata, cluster, and weight according to the guidelines for using KYRBWS raw data. A *t*-test and One-way ANOVA were used to analyze the general characteristics, types, levels of physical activity, and mental health. If there was a significant difference in the repeated-measures ANOVA, the post hoc Scheffé’s test confirmed the interaction differences between the groups. In addition, we analyzed the differences in suicidal behavior factors according to general characteristics and physical activity type and level using the chi-square test. To analyze the effects of the type and frequency of physical activity on mental health and suicidal behavior in adolescents, we controlled for confounding variables (gender, school type, grade, region scale, and region) that could potentially affect the results. The significance level for all statistical analyses was established at 0.05 (*p* < 0.05).

## 3. Results

Table 1 presents the differences in the types of physical activity (>60 min PA, Moderate PA, Vigorous PA, Strength training, walking, PE class PA, and PA on the move) and frequency (number of days/week) according to the general characteristics of the adolescents (gender, school type, grade, region scale, and region). In terms of gender, grade, school type, and region, there was a significant difference according to all types of physical activity (*p* < 0.05, *p* < 0.01, *p* < 0.001), and at the regional scale, there was a significant difference in the level of physical activity in Moderate PA, Walks, PE class PA, and PA on the move (*p* < 0.05, *p* < 0.01, *p* < 0.001) in the case of statistically significant differences, post hoc analysis was performed, and males showed higher PA levels than females, but females showed higher PA levels than males only in PA. Girls’ schools showed the lowest levels of all PA types. The grades showed that the level of most PA types decreased as the grades increased, and the level of PA types was higher in middle school students than in high school students. At the regional scale, metropolises showed high PA levels, and counties showed low PA levels.

Table 2 presents the level of self-rated perceived stress (loneliness) according to the general characteristics of the adolescents; there were significant differences in all factors (*p* < 0.05, *p* < 0.01, *p* < 0.001). If there was a statistically significant difference, a post hoc test was conducted. In terms of gender, males showed more positive mental health than females, and boys mixed with girls showed more positive mental health. The 7th grade showed that the mental health level was negative, as the grade was the highest. Location was found to have the most negative effect on mental health in small and medium-sized cities. In this region, Youngnam showed the most positive mental health factor.

Table 3 presents the experience of suicidal behavior factors (suicidal ideation, plan, attitude, and hopelessness) according to the general characteristics of adolescents; there were significant differences in all factors except for hopelessness on the regional scale and suicidal attitudes in the region (*p* < 0.05, *p* < 0.01, *p* < 0.001). Regarding sex, females had more experience with suicidal behavior factors than males, and in terms of school type, the mixed group had more experience with suicidal behavior factors than boys and girls. Students in grades 7th–9th (middle school students) showed higher experience levels of suicidal behavior factors than those in 10th–12th (high school students). Small/medium cities were higher than in other locations, and the experience rate was lowest among county-resident adolescents. The experience of suicidal behavior factors was highest in the capital city and lowest in Jeju. Suicidal ideation was experienced by 14.1% (*n* = 7297), suicidal planning was experienced by 4.5% (*n* = 2301), suicidal attitude was experienced by 2.7% (*n* = 1376), and hopelessness was experienced by 16.6% (*n* = 14,880).

Table 4 presents the difference in the level of mental health problems (self-rated health, perceived stress, and loneliness) according to the type and frequency of PA, and there was a statistically significant difference in all factors (*p* < 0.05, *p* < 0.001), and post hoc analyses were conducted. Self-rated health was positive as the frequency of >60 min PA, Moderate PA, Vigorous PA, Strength training, and PE class PA increased, and Walks and PA on the move were also positive as the frequency of PA increased. Perceived stress was the most negative in all types of PA when the frequency was 0 days/week. Loneliness was found to be less frequent as the frequency of most PA types increased, but the level of loneliness was lower as the frequency of PA was lower in Walks and PA on the move.

Table 5 presents the number of experiences of suicidal behavior factors (suicidal ideation, suicidal plan, suicidal attempt, hopelessness) according to the type and frequency of PA, and there was a statistically significant difference in the frequency of >60 min PA and PA on the move, suicidal ideation, and the frequency of Strength training except for the relationship with hopelessness (*p* < 0.05, *p* < 0.01, *p* < 0.001).

Suicidal ideation decreased with increasing frequencies of Moderate PA, Vigorous PA, and Strength training, whereas with walking it increased. The PE class showed the lowest experience rate at a frequency of 1–2 days/week, but there was no significant difference between the frequency of >60 min PA and PA on move. The frequency of suicidal plans was lower at >60 min of PA, Moderate PA, Vigorous PA, and with Strength training, frequency increased. On the other hand, walking, PE class PA, and PA on the move showed the lowest experience rate at a 1–2 days/week frequency. The suicidal attitude frequency was lower, with a frequency of >60 min PA, moderate PA, and Vigorous PA, and increased Strength training physical activity. For Walks, PE class PA, and PA on the move, the experience rate was low, at a frequency of 1–2 days/week. Finally, hopelessness had a lower experience rate, with a frequency of >60 min of PA, Moderate PA, and Vigorous PA. PE class PA and PA on the move showed the lowest hopelessness experience rate at 1–2 days/week PA frequency. However, the experienced rate of hopelessness increased as the frequency of walking increased, and the frequency of Strength training was not significantly related to hopelessness.

## 4. Discussion

This study investigated the levels of physical activity, mental health, and suicidal behaviors among Korean adolescents. In addition, by identifying the relationship between mental health and suicidal behavior according to the level of PA, it is necessary to present guidelines for the type and frequency of Pas that are practically suitable for Korean adolescents. The main results of this study and the discussion about them are as follows.

As adolescents grow and develop, mental health and suicidal behavior problems are caused by a complex interaction of social, environmental, and genetic factors [30,31]. Recognition and work to overcome challenges that arise during the complex interactions that adolescents experience are referred to as adaptation and resilience [32,33]. In other words, adaptation and resilience are necessary factors for coping with the constantly changing environment and relationships of adolescence [34]. Therefore, when discussing the differences in mental health, suicidal behavior, and physical activity among Korean adolescents obtained from the results of this study, we will analyze the situation of Korean adolescents and compare the characteristics of socioenvironmental and genetic factors by observing other countries’ cases.

According to the general characteristics of the adolescents, the number of PAs per week was significantly higher in males than in females. This trend was also reflected in the lowest frequency of all: PA types among adolescents attending girls’ schools compared to boys’ schools and coeducational schools. These findings are common not only in Korean girls but also in other countries (36 countries) [35], and the participation level in MVPA was significantly low [36]. In general, female adolescents prefer passive activities to males [37] and have socioenvironmental characteristics to form relationships within familiar communities, such as family at home or friends at school [36]. In addition, parents can see that low PA levels continue into adolescence because of the concern that a girl may be injured during PA [38].

In terms of grade and PA, middle school students (7th-9th grade) showed a higher frequency of PA than high school students (10th-12th grade) and tended to decrease as the grade increased; 12th grade showed the lowest frequency of all types of PA. This seems to be because Korean youth are so enthusiastic about their studies that they give up other parts of their lives to enter university [29]. However, in the case of American adolescents, where competition for entrance examinations is important, the level of PA tends to decrease in the 9th-11th grade. Nonetheless, it increases slightly in the 12th grade [39]. In the case of Chinese adolescents, the weekly physical activity (MVPA, VPA, MPA, walking) time increases as the grade increases [40]. This suggests that the difference in the level of physical activity by grade can vary depending on the social environment and the direction pursued by each country, and it is evident that it is necessary to find ways to increase the level of physical activity even if the grade of Korean adolescents increases.

At the PA level, according to the Korean adolescent region scale, the metropolitan area was the highest in MPA, walking, PE class PA, and PA on the move, and the PA level of the rural area was the lowest. In addition, when the residential area of adolescents was subdivided into regions, the capital showed the highest tendency, and the local area showed a low level of PA. However, the levels of the physical activity of Chinese and British Scottish adolescents in urban and rural areas were similar or slightly higher in terms of fat [41,42]. Previous studies have shown that the level of physical activity naturally increases in rural areas due to a nature-friendly environment [43]. However, in Korea, it seems that the physical activity level is higher than that of the local area because the physical facility space where youth can perform physical activity is sufficiently secured in metropolitan areas. 

In terms of mental health and suicidal behavior, according to the general characteristics of adolescents, female adolescents were more negative for all factors of mental health (self-rated health, perceived stress, and loneliness) and suicidal behavior (suicidal ideation, suicidal plan, suicidal attitude, and hopelessness) than male adolescents. Because of the negative responses to mental health and suicidal behavior among female adolescents, it seems that there are also negative aspects of mental health and suicidal behavior in the school type (Girls’, mixed school) that female adolescents attend. The level of life satisfaction (psychological distress) among adolescents worldwide (73 countries, 566,829 adolescents) was also higher in male adolescents than in female adolescents [44], and the rate of suicidal behavior was higher in female adolescents [45]. In the case of female adolescents, there is a possibility that they will experience mental health problems such as depression quickly while entering physical changes such as puberty faster than men. Male adolescents engage in a variety of external activities to cope with stress, but female adolescents may experience physical damage rather than emotional problems [46]. Therefore, there is a need for a solution to lower the risk of mental health problems and suicidal behavior and to express emotional problems in female adolescents. In another study, parental support was reported to be significantly associated with mental health and suicidal behavior in Korean adolescents [32]. In particular, Korean female students were found to be more sensitive to stressful life events when their parents’ socioeconomic support was weaker, resulting in higher levels of depression and problem behavior [33]. The National Youth Policy Report attributed this to the fact that female adolescents from poor families begin economic activities at an early age and experience stress when they have to give up entering college [47]. In addition, parents’ emotional support (trust, affection, respect, and understanding) is also a factor in reducing suicidal ideation through psychological stability for Korean female adolescents, which is also because female adolescents are highly dependent on their parents [33,48]. Therefore, it can be interpreted that adequate parental support is necessary to reduce the risk of mental health and suicidal behavior among female adolescents in Korea.

In terms of mental health according to grade, the higher the grade, the more negatively students responded. In terms of perceived stress, high school students experienced more stress than middle school students. California adolescents in the United States showed the same results, as their mental health levels decreased with an increase in grade [49]. One of the causes that negatively affects the level of stress and mental health of adolescents in high school may be the academic stress of entering university. However, the social and environmental factors of school life, family, and friendships have an impact. However, the experience of suicidal behavioral factors was higher among middle school students than among high school students. The middle school period can be attributed to the fact that it is difficult to adapt to various hormonal and physical changes and rapidly changing social and environmental factors since the beginning of adolescence biologically [46,50].

The mental health of adolescents in areas such as small and medium cities, the Capital, and Chungcheong was found to be the most negative. Previous studies have shown that the more developed adolescent residential areas are, the more likely they are to have mental health problems, and there is no significant difference [51,52]. However, when the economic gap is greater in the environment in which they live than in the development level of the residential area, it affects the mental health of youths [52]. Therefore, in the case of small/medium cities in Korea, these have been developed to some extent. However, since undeveloped areas still exist, it can be seen that adolescents’ low mental health levels have appeared. In contrast, the Youngnam area was the lowest in the location and highest in the rest of the area. It is necessary to review and implement various supports and national policies to solve the mental health problems of adolescents according to their residential area [53].

The experience of suicidal behavior according to the residential area of adolescents was also the highest in small/medium cities and the capital. In the US, the smaller the residential area, the greater the number of adolescents visiting the emergency room due to suicidal ideation and self-harm [54]. Previous studies have shown that the relationship between adolescents’ suicidal behavior experiences and residential areas is of low economic income [52] and pointed out that there are not enough facilities to provide appropriate treatment until adolescents’ suicidal behavior is experienced [55,56]. However, it is urgent to identify the cause of the current situation and prepare countermeasures against it, because Korea has shown that suicidal behavior experiences are higher in large cities and small/medium city adolescents who have developed.

The level of adolescents’ mental health-related suicidal behavior experience according to the type of physical activity and frequency of participation they follow is as follows: Mental health and suicidal behavior experiences according to the frequency of PA > 60 min were most negative at 0 days/week. Self-rated health improved as the frequency of PA increased. Perceived stress was positive when it was over three days/week, and loneliness was significantly effective when it was over one day/week. In the case of suicidal behavior experiences, there was no significant difference between the frequency of this and of suicidal ideation. However, for the other factors (suicidal plans, suicide attempts, and hopelessness), the rate of suicidal behavior experience decreased as the frequency of PA increased. >PA for 60 min of PA is known to affect mental health, such as stress, anxiety, and depression [57]. Adolescents in the US were 2.18 times more likely to experience depression if they did not perform > 60 min PA [58], and suicidal ideation decreased in both men and women if they performed > 60 min of PA more than once a week [59]. Therefore, if adolescents perform > 60 min of PA more than once a week, it will positively affect their mental health and suicidal behavior. Based on the results of this study, the recommended frequency of physical activity > 60 min PA, which is effective for mental health and suicidal behavior in Korean adolescents, is >3 days/week.

The frequency of physical activity in MPA and VPA(MVPA) showed the same effect on self-rated health and perceived stress on mental health. Self-rated health increased as the frequency increased, and perceived stress was most effective when over three days/week. MPA was not significantly different in the post hoc test of loneliness, but 5–7 days/week of VPA was more effective than the other PAs. In addition, as the frequency of MVPA increased, the experience rate decreased. However, in the case of hopelessness, the frequency of 1–2 days/week was higher than that of 0 days/week, but it was confirmed that the experience rate of over 3 days/week was lower than that of 0 days/week. Swedish adolescents who perform MVPA during their leisure time have reduced anxiety and mental health-related quality of life (QoL) [60]. In addition, MVPA was found to significantly lower the risk of mental illness and suicidal behavior compared to low physical activity (LPA) levels in adolescents [61,62]. Therefore, MVPA frequency, which is effective for adolescents’ mental health and suicidal behavior experience, is over 3 days/week, and it is necessary to encourage adolescents to participate in MVPA more actively than other types of PA.

Strength training was 0 days/week and was found to have the most negative effect on all factors of mental health and suicidal behavior. Self-rated health improved as the frequency of Strength training increased, and perceived stress was more effective at 5–7 days/week than at 1–4 days/week. Loneliness was found to have a greater effect for 5–7 days/week than for 1–2 days/week. Suicidal behavior decreased as the frequency of Strength training increased, but there was no significant difference in the frequency of 3–4 and 5–7 days/week. Regardless of the exercise method of Strength training, one study found a positive effect on anxiety and depression in adolescents [63]. However, there is a lack of research on the direct effects of Strength training on adolescent suicidal behaviors. Studies have shown that the higher the handgrip strength of adults, the lower their suicidal thoughts [64]. Therefore, adolescents’ Strength training can reduce the risk of experiencing suicidal behavior, and the recommended frequency of PA effective for adolescents’ mental health and suicidal behavior is over 3 days/week.

Walking was found to have the most negative impact on self-rated health and perceived stress among mental health factors at 0 days/week. Self-rated health was most effective at 5–7 days/week, and perceived stress did not differ over 1 day/week. Loneliness in mental health and suicidal behavior experiences showed negative results as the frequency of walking increased. In particular, a frequency of 5–7 days/week caused the most adverse effects. Other studies have reported walking for 60 min/week minutes/week has a positive effect on adolescents’ mental health [65]. Additionally, the walking practice of 60 min/week was similar to MVPA in adolescents’ suicidal behavior experience rates [66]. However, in this study, the frequency of 5–7 days/week (50–70 min/week) was positive only for some mental health factors, while the rest showed negative effects. Therefore, Korean adolescents need additional recommendations for other types of PA rather than walking as a single exercise.

As the frequency of PA in the PE class increased, the positive effects on self-rated health and perceived stress increased. Loneliness also had a greater effect for over 3 days/week than 1–2 days/week. However, it is more important for mental health and cognitive function to perform PA in Physical education classes by increasing the intensity rather than the participation frequency for adolescents [67]. To lower the risk of mental health and suicidal behavior in adolescents, the frequency of PE-class PA can be sufficiently high at 1 day/week level, but it is necessary to induce students to increase their exercise intensity during PE class.

Among the frequencies of PA, 0 days/week was negative for self-rated health and one’s perceived stress of mental health. Self-rated health had a similar effect on the frequency of over 1 day/week, and perceived stress had a greater effect on 1–2 and 5–7 days/week than on day 0. Regarding loneliness, the frequency of 3–7 days/week was more negative than that of zero days/week. Regarding suicidal behavior, the highest experience rate was observed in the frequency of 5–7 days/week, and the lowest experience rate was observed in the frequency of 1–4 days/week. Although physical activities such as going to school and playing with friends are classified as MVPA [22], the results of PA on the move in this study are different from those of MVPA. Therefore, it can be expected that the physical activity intensity of PA on the movement of Korean adolescents is lower than that of MVAP. However, when adolescents (males and females) from 34 countries participated in physical activities such as walking or bicycling while attending school, their suicidal attitudes were significantly lowered [68]. Therefore, to observe the effect of PA on the mental health and suicidal behavior of adolescents, it is necessary to encourage other types of PA while carrying out physical activities while moving.

In this study, we found that the risk of mental health and suicidal behavior decreased as Korean adolescents participated in higher-intensity physical activities than LPA (Walks and PA on move). However, mental health and suicidal behavior outcomes in adolescents have complex background characteristics. Based on these findings, it is recommended that adolescents be encouraged to engage in more frequent physical activity of a certain intensity, but additional interventions that take into account the sociocultural and specific needs of the individual may further reduce the risk of mental health and suicidal behavior.

This study presents meaningful results on the types and levels of physical activity (frequency) required to effectively lower the risk of mental health and suicidal behavior in Korean adolescents. However, there are some limitations. First, it comprehensively suggests the types of PA in which adolescents can participate. Second, factors related to the level of PA in adolescents were included only in the type and frequency, and no other variables (time, volume, and intensity) were analyzed. Therefore, it is necessary to analyze the relationship between mental health and suicidal behavior factors, including various variables that can be used to classify physical activity types and classify physical activity levels. Follow-up studies are necessary to present practical and specific physical activity standards that can lower the risk of mental health problems and suicidal behaviors among Korean adolescents. Third, the data collection method utilized in this study is cross-sectional and uses secondary data variables, which limits the ability to closely identify causal relationships between specific variables in detail. Therefore, future research should include longitudinal studies that explore causal factors that take into account multiple variables related to mental health and suicidal behavior in adolescents. Fourth, the survey in this study a self-report survey. This survey method is subject to potential response bias, which could affect the results of the study. Fifth, the mental health and suicidal behavior variables used in this study consisted of single-item questions, which compromise the validity of the assessment tool. Further research is needed to confirm the relationship between mental health and suicidal behavior among Korean adolescents using assessment tools with validity and reliability.

## 5. Conclusions

Through the results of this study, the PA level, mental health, and suicidal behavior experience rates of Korean adolescents by type were identified, and the relationship between PA type and level was analyzed. Among the general characteristics of Korean adolescents, the level of PA by gender was higher in males than in females and schools attended by male adolescents (mixed and Boys’). One of the ways to increase PA levels in female adolescents is to actively engage them in physical activity through after-school programs and sports clubs where they can experience and enjoy different types of exercise. The higher the grade of adolescents, the lower the level of physical activity, and adolescents in the level of physical activity was higher in the metropolitan area than in other regions. To address the disparity in PA level across regions, it will be necessary to build physical activity infrastructure such as gyms and parks in areas that lack Physical education facilities. Mental health was worse in female adolescents, lower in the upper grades, and in rural areas. Suicidal behavior was higher in female adolescents, but the suicidal experience rate was lower in the upper grades. Small/medium city sizes and metropolitan areas had the highest suicide rates. In terms of PA type and mental health level, regardless of the type of PA, the higher the participation frequency, the more positive the mental health level; however, loneliness was negative as the frequency of walking and PA on the move increased. In terms of the relationship between the types of PA and suicidal behavior, the rate of suicidal behavior decreased as the frequency of over 60 min of PA, MPA, VPA, and Strength training increased. However, the rate of suicidal behavior was the lowest at 1–2 days/week for Walks and PA on the move, and 1 day/week for PE class PA. Based on the results of this study, it is effective for Korean adolescents to increase their PA level (frequency), regardless of their type, to lower their risk of mental health problems and suicidal behavior as a whole. However, in some types of physical activity, such as LPA (Walks, PA on the move) and PE class PA, the effect did not reduce the risk of mental health and suicidal behavior, even if the frequency increased. Therefore, efforts should be made to increase physical activity levels among Korean adolescents. In particular, recommendations and policies are needed to create social environmental factors and cultures that can perform physical activities at a certain level or more, such as over 60 min of PA, MPA, VPA, and Strength training. However, there are potential risks of overtraining while participating in PA, such as physical injury and the exacerbation of mental health issues, so careful consideration should be given to determining the type and level of physical activity that is appropriate for adolescents. Furthermore, the complex relationship between mental health and suicidal behavior needs to be recognized and approached when conducting research on developing policies and programs that are appropriate to the individual differences, specific needs, socioenvironmental factors, and culture of adolescents.

## Figures and Tables

**Table 1 behavsci-14-00160-t001:** Types and levels of physical activity according to general characteristics.

Group	Participants (n = 51,636)		Number of Days/Week						
			>60 min PA	Moderate PA	Vigorous PA	Strength Training	Walks	PE Class PA	PA on the Move
	Variables	% (n)	M ± SD	M ± SD	M ± SD	M ± SD	M ± SD	M ± SD	M ± SD
Gender	Male	50.9 (26,257)	3.68 ± 2.29	3.65 ± 2.26	3.67 ± 2.23	3.42 ± 2.76	6.62 ± 2.05	2.86 ± 1.09	4.78 ± 3.00
	Female	49.1 (25,379)	2.60 ± 1.86	2.57 ± 1.85	2.97 ± 1.95	1.76 ± 1.62	6.56 ± 2.00	2.71 ± 1.10	4.83 ± 2.96
*t*			558.981 ***	59.604 ***	38.106 ***	84.053 ***	3.450 **	15.528 ***	−2.041 *
School type	Mixed	68.5 (35,385)	3.21 ± 2.19	3.38 ± 2.14	3.18 ± 2.19	2.66 ± 2.46	6.60 ± 2.02	2.85 ± 1.10	4.86 ± 2.98
	Boys’	15.7 (8129)	3.54 ± 2.20	3.55 ± 2.20	3.51 ± 2.18	3.27 ± 2.68	6.65 ± 2.05	2.76 ± 1.06	4.79 ± 2.99
	Girls’	15.7 (8122)	2.52 ± 1.83	2.87 ± 1.94	2.48 ± 1.81	1.68 ± 1.51	6.50 ± 2.03	2.56 ± 1.09	4.57 ± 2.96
*F*			500.938 ***	251.215 ***	521.183 ***	943.476 ***	11.868 ***	224.923 ***	29.830 ***
Post hoc			C < A < B	C < A < B	C < A < B	C < A < B	C < A < B	C < B < A	C < B < A
Grade	7th grade	17.8 (9205)	3.42 ± 2.20	3.63 ± 2.09	3.47 ± 2.12	2.68 ± 2.37	6.51 ± 2.01	3.04 ± 1.07	4.97 ± 2.88
	8th grade	18.0 (9316)	3.46 ± 2.22	3.66 ± 2.14	3.49 ± 2.18	2.72 ± 2.45	6.62 ± 1.99	3.21 ± 1.06	5.07 ± 2.92
	9th grade	18.2 (9387)	3.38 ± 2.24	3.66 ± 2.16	3.40 ± 2.20	2.68 ± 2.49	6.74 ± 1.90	3.20 ± 1.08	5.18 ± 2.91
	10th grade	16.3 (8424)	2.89 ± 2.02	3.10 ± 2.03	2.86 ± 2.01	2.54 ± 2.39	6.56 ± 2.08	2.61 ± 0.89	4.62 ± 3.04
	11th grade	15.4 (7956)	2.94 ± 2.06	3.03 ± 2.05	2.84 ± 2.04	2.56 ± 2.42	6.60 ± 2.04	2.48 ± 0.93	4.51 ± 3.03
	12th grade	14.2 (7348)	2.67 ± 2.04	2.70 ± 2.08	2.49 ± 2.01	2.37 ± 2.35	6.49 ± 2.13	1.97 ± 0.96	4.29 ± 3.04
*F*			203.729 ***	312.606 ***	327.804 ***	23.048 ***	17.675 ***	1937.353 ***	119.266 ***
Post hoc			F < DE < ABC	F < DE < ABC	F < DE < ABC	F < DE < ABC	AF < B < C/ F < DE < C	F < E < D < A < BC	F < DE < AB < C
Location	County	7.4 (3812)	3.20 ± 2.14	3.28 ± 2.08	3.17 ± 2.14	2.63 ± 2.42	6.11 ± 2.25	2.75 ± 1.11	4.25 ± 2.97
	Metropolis	42.8 (22,121)	3.16 ± 2.16	3.36 ± 2.14	3.14 ± 2.12	2.58 ± 2.41	6.73 ± 1.94	2.80 ± 1.09	5.01 ± 2.96
	Small/medium city	49.8 (25,703)	3.14 ± 2.16	3.31 ± 2.12	3.10 ± 2.14	2.62 ± 2.43	6.55 ± 2.04	2.78 ± 1.10	4.71 ± 2.98
*F*			1.256	5.664 **	2.726	1.360	166.029 ***	3.632 *	134.060 ***
Post hoc				C < B			A < C < B	A < B	A < C < B
Region	Capital	42.0 (21,671)	3.13 ± 2.15	3.37 ± 2.14	3.12 ± 2.13	2.57 ± 2.40	6.73 ± 1.94	2.82 ± 1.08	5.01 ± 2.95
	Kangwon	3.7 (1927)	3.12 ± 2.15	3.18 ± 2.03	3.11 ± 2.08	2.61 ± 2.43	6.29 ± 2.12	2.78 ± 1.07	4.19 ± 2.94
	Chungcheong	13.5 (6963)	3.15 ± 2.12	3.28 ± 2.10	3.08 ± 2.10	2.61 ± 2.43	6.42 ± 2.09	2.75 ± 1.10	4.66 ± 2.96
	Honam	12.4 (6400)	3.10 ± 2.14	3.24 ± 2.11	3.05 ± 2.13	2.61 ± 2.42	6.41 ± 2.14	2.73 ± 1.13	4.50 ± 3.02
	Youngnam	25.9 (13,369)	3.21 ± 2.20	3.36 ± 2.14	3.18 ± 2.16	2.66 ± 2.44	6.60 ± 2.03	2.79 ± 1.11	4.85 ± 2.99
	Jeju	2.5 (1306)	3.29 ± 2.16	3.33 ± 2.13	3.21 ± 2.17	2.52 ± 2.30	6.41 ± 2.11	2.79 ± 1.14	4.00 ± 2.95
*F*			4.380 **	6.279 ***	4.049 **	3.056 **	52.685 ***	8.061 ***	74.328 ***
Post hoc				BD < A/D < E	D < E	A < E	BCD < E < A/F < A	CD < A	BF < CD < E < A

Notes: PA = physical activity; PE = physical education; * *p* < 0.05; ** *p* < 0.01; *** *p* < 0.001.

**Table 2 behavsci-14-00160-t002:** Self-rated health, perceived stress, and loneliness experience according to general characteristics.

Group	Participants (n = 51,636)		Self-Rated Health	Perceived Stress	Loneliness
	Variables	% (n)	M ± SD	M ± SD	M ± SD
Gender	Male	50.9 (26,257)	2.16 ± 0.93	2.79 ± 0.97	2.37 ± 1.09
	Female	49.1 (25,379)	2.39 ± 0.89	2.54 ± 0.93	2.71 ± 1.07
*T*			−29.157 ***	30.355 ***	−36.252 ***
School type	Mixed (A)	68.5 (35,385)	2.27 ± 0.91	2.66 ± 0.96	2.56 ± 1.09
	Boys’ (B)	15.7 (8129)	2.16 ± 0.93	2.80 ± 0.96	2.33 ± 1.08
	Girls’ (C)	15.7 (8122)	2.41 ± 0.89	2.53 ± 0.93	2.65 ± 1.08
*F*			160.635 ***	156.381 ***	204.496 ***
Post hoc			B < A < C	C < A < B	B < A < C
Grade	7th grade (A)	17.8 (9205)	2.15 ± 0.84	2.76 ± 0.95	2.46 ± 1.08
	8th grade (B)	18.0 (9316)	2.23 ± 0.90	2.69 ± 0.95	2.53 ± 1.09
	9th grade (C)	18.2 (9387)	2.27 ± 0.92	2.66 ± 0.95	2.57 ± 1.09
	10th grade (D)	16.3 (8424)	2.32 ± 0.92	2.63 ± 0.96	2.61 ± 1.08
	11th grade (E)	15.4 (7956)	2.33 ± 0.94	2.63 ± 0.97	2.56 ± 1.10
	12th grade (F)	14.2 (7348)	2.37 ± 0.98	2.59 ± 0.99	2.51 ± 1.11
*F*			66.321 ***	31.321 ***	20.885 ***
Post hoc			A < BC < DE < F	DEF < BC < A	AF < BCDE
Location	County (A)	7.4 (3812)	2.25 ± 0.96	2.71 ± 0.96	2.50 ± 1.10
	Metropolitan (B)	42.8 (22,121)	2.26 ± 0.92	2.66 ± 0.96	2.53 ± 1.09
	Small/medium city (C)	49.8 (25,703)	2.29 ± 0.91	2.66 ± 0.96	2.55 ± 1.09
*F*			5.640 **	5.675 **	3.931 *
Post hoc			B < C	BC < A	A < C
Region	Capital (A)	42.0 (21,671)	2.29 ± 0.92	2.63 ± 0.96	2.59 ± 1.09
	Kangwon (B)	3.7 (1927)	2.31 ± 0.92	2.69 ± 0.94	2.52 ± 1.08
	Chungcheong (C)	13.5 (6963)	2.29 ± 0.91	2.65 ± 0.97	2.55 ± 1.11
	Honam (D)	12.4 (6400)	2.30 ± 0.91	2.65 ± 0.95	2.51 ± 1.08
	Youngnam (E)	25.9 (13,369)	2.22 ± 0.91	2.72 ± 0.96	2.47 ± 1.08
	Jeju (F)	2.5 (1306)	2.30 ± 0.93	2.71 ± 0.98	2.54 ± 1.11
*F*			11.179 ***	16.767 ***	23.147 ***
Post hoc			E < ABCD	ACD < E	DE < AC

Notes: * *p* < 0.05; ** *p* < 0.01; *** *p* < 0.001.

**Table 3 behavsci-14-00160-t003:** Suicidal ideation, plan, attempt, and hopelessness experience according to general characteristics.

Group	Participants (n = 51,636)		Suicidal Ideation		Suicidal Plan		Suicidal Attempt		Hopelessness	
	Variables	% (n)	No % (n)	Yes % (n)	No % (n)	Yes % (n)	No % (n)	Yes % (n)	No % (n)	Yes % (n)
Gender	Male	50.9 (26,257)	45.5 (23,477)	5.4 (2780)	49.0 (25,311)	1.8 (946)	49.9 (25,743)	1.0 (514)	38.6 (19,938)	12.2 (6319)
	Female	49.1 (25,379)	40.4 (20,862)	8.7 (4517)	46.5 (24,024)	2.6 (1355)	47.5 (24,517)	1.7 (862)	32.6 (16,818)	16.6 (8561)
χ^2^			552.937 ***		91.370 ***		103.018 ***		587.886 ***	
School type	Mixed	68.5 (35,385)	58.5 (30,220)	10.0 (5165)	65.3 (33,717)	3.2 (1668)	66.6 (34,374)	2.0 (1011)	48.2 (24,864)	20.4 (10,521)
	Boys’	15.7 (8129)	14.2 (7344)	1.5 (785)	15.2 (7867)	0.5 (262)	15.5 (8004)	0.2 (125)	12.2 (6321)	3.5 (1808)
	Girls’	15.7 (8122)	13.1 (6775)	2.6 (1347)	15.0 (7751)	0.7 (371)	15.3 (7882)	0.5 (240)	10.8 (5571)	4.9 (2551)
χ^2^			180.720 ***		34.790 ***		47.496 ***		212.415 ***	
Grade	7th grade	17.8 (9205)	15.2 (7837)	2.6 (1368)	17.0 (8779)	0.8 (426)	17.3 (8917)	0.6 (288)	13.1 (6764)	4.7 (2441)
	8th grade	18.0 (9316)	15.2 (7863)	2.8 (1453)	17.1 (8855)	0.9 (461)	17.5 (9046)	0.5 (270)	12.8 (6629)	5.2 (2687)
	9th grade	18.2 (9387)	15.5 (7995)	2.7 (1392)	17.3 (8943)	0.9 (444)	17.7 (9114)	0.5 (273)	12.8 (6613)	5.4 (2774)
	10th grade	16.3 (8424)	14.1 (7306)	2.2 (1118)	15.7 (8127)	0.6 (297)	16.0 (8243)	0.5 (181)	11.5 (5920)	4.8 (2504)
	11th grade	15.4 (7956)	13.4 (6901)	2.0 (1055)	14.7 (7611)	0.7 (345)	15.0 (7763)	0.4 (193)	10.9 (5643)	4.5 (2313)
	12th grade	14.2 (7348)	12.5 (6437)	1.8 (911)	13.6 (7020)	0.6 (328)	13.9 (7177)	0.3 (171)	10.0 (5187)	4.2 (2161)
χ^2^			52.596 ***		24.995 ***		25.375 ***		31.078 ***	
Location	County	7.4 (3812)	6.4 (3323)	0.9 (489)	7.1 (3651)	0.3 (161)	7.2 (3717)	0.2 (95)	5.3 (2736)	2.1 (1076)
	Metropolitan	42.8 (22,121)	36.9 (19,073)	5.9 (3048)	41.0 (21,194)	1.8 (927)	41.8 (21,580)	1.0 (541)	30.5 (15,769)	12.3 (6352)
	Small/medium city	49.8 (25,703)	42.5 (21,943)	7.3 (3760)	47.4 (24,490)	2.3 (1213)	48.3 (24,963)	1.4 (740)	35.3 (18,251)	14.4 (7452)
χ^2^			12.842 **		8.329 *		9.083 *		1.147	
Region	Capital	42.0 (21,671)	35.5 (18,319)	6.5 (3352)	39.9 (20,600)	2.1 (1071)	40.8 (21,063)	1.2 (608)	29.4 (15,198)	12.5 (6473)
	Kangwon	3.7 (1927)	3.3 (1680)	0.5 (247)	3.6 (1850)	0.1 (77)	3.6 (1875)	0.1 (52)	2.7 (1387)	1.0 (540)
	Chungcheong	13.5 (6963)	11.6 (5992)	1.9 (971)	12.9 (6678)	0.6 (285)	13.1 (6790)	0.3 (173)	9.5 (4923)	4.0 (2040)
	Honam	12.4 (6400)	10.7 (5519)	1.7 (881)	11.8 (6115)	0.6 (285)	12.1 (6238)	0.3 (162)	8.8 (4541)	3.6 (1859)
	Youngnam	25.9 (13,369)	22.7 (11,710)	3.2 (1659)	24.9 (12,849)	1.0 (520)	25.2 (13,030)	0.7 (339)	18.9 (9758)	7.0 (3611)
	Jeju	2.5 (1306)	2.2 (1119)	0.4 (187)	2.4 (1243)	0.1 (63)	2.4 (1264)	0.1 (42)	1.8 (949)	0.7 (357)
χ^2^			68.245 ***		25.628 ***		5.365		35.915 ***	
Total			85.9 (44,339)	14.1 (7297)	95.5 (49,335)	4.5 (2301)	97.3 (50,260)	2.7 (1376)	71.2 (36,756)	16.6 (14,880)

Notes: * *p* < 0.05; ** *p* < 0.01; *** *p* < 0.001.

**Table 4 behavsci-14-00160-t004:** Self-rated health, perceived stress, and loneliness according to types and levels of PA.

Type	Participants (n = 51,636)		Self-Rated Health	Perceived Stress	Loneliness
	Frequency	% (n)	M ± SD	M ± SD	M ± SD
> 60 min PA	0 (A)	33.1 (17,075)	2.43 ± 0.93	2.63 ± 0.97	2.54 ± 1.11
	1–2 (B)	29.1 (15,028)	2.34 ± 0.88	2.65 ± 0.92	2.57 ± 1.06
	3–4 (C)	21.1 (10,888)	2.19 ± 0.89	2.70 ± 0.95	2.54 ± 1.07
	5–7 (D)	16.7 (8645)	1.96 ± 0.91	2.73 ± 1.00	2.49 ± 1.13
*F*			569.250 ***	28.504 ***	7.983 ***
Post hoc			D < C < B < A	AB < CD	D < ABC
Moderate PA	0 (A)	27.1 (13,972)	2.45 ± 0.96	2.58 ± 1.00	2.52 ± 1.14
	1–2 (B)	32.2 (16,652)	2.31 ± 0.87	2.67 ± 0.92	2.55 ± 1.06
	3–4 (C)	22.8 (11,771)	2.20 ± 0.89	2.71 ± 0.93	2.56 ± 1.07
	5–7 (D)	17.9 (9241)	2.05 ± 0.91	2.72 ± 1.00	2.52 ± 1.12
*F*			415.750 ***	52.129 ***	3.176 *
Post hoc			D < C < B < A	A < B < CD	
Vigorous PA	0 (A)	34.4 (17,739)	2.47 ± 0.94	2.59 ± 0.98	2.55 ± 1.12
	1–2 (B)	28.0 (14,443)	2.31 ± 0.88	2.67 ± 0.93	2.56 ± 1.06
	3–4 (C)	20.9 (10,798)	2.17 ± 0.87	2.72 ± 0.94	2.55 ± 1.07
	5–7 (D)	16.8 (8656)	1.94 ± 0.89	2.75 ± 0.99	2.47 ± 1.12
*F*			746.973 ***	97.001 ***	13.249 ***
Post hoc			D < C < B < A	A < B < CD	D < ABC
Strength training	0 (A)	54.5 (28,136)	2.42 ± 0.91	2.61 ± 0.95	2.56 ± 1.09
	1–2 (B)	21.1 (10,904)	2.22 ± 0.88	2.71 ± 0.94	2.53 ± 1.07
	3–4 (C)	11.5 (5915)	2.09 ± 0.89	2.72 ± 0.95	2.50 ± 1.09
	5–7 (D)	12.9 (6681)	1.91 ± 0.89	2.78 ± 1.01	2.48 ± 1.13
*F*			721.402 ***	83.522 ***	14.305 ***
Post hoc			D < C < B < A	A < BC < D	CD < A/D < B
Walks	0 (A)	4.2 (2161)	2.45 ± 1.02	2.54 ± 1.09	2.40 ± 1.21
	1–2 (B)	7.0 (3594)	2.39 ± 0.93	2.66 ± 0.97	2.49 ± 1.08
	3–4 (C)	10.2 (5282)	2.35 ± 0.90	2.67 ± 0.94	2.55 ± 1.07
	5–7 (D)	78.6 (40,599)	2.24 ± 0.91	2.67 ± 0.95	2.55 ± 1.09
*F*			72.145 ***	12.401 ***	15.265 ***
Post hoc			D < C < A/D < B	A < BCD	A < B < D/A < C
PE class PA	0 (A)	18.7 (9655)	2.47 ± 0.97	2.55 ± 1.00	2.54 ± 1.15
	1 (B)	16.7 (8624)	2.35 ± 0.92	2.64 ± 0.96	2.55 ± 1.09
	2 (C)	31.7 (16,349)	2.27 ± 0.89	2.68 ± 0.94	2.56 ± 1.07
	Over 3 (D)	32.9 (17,008)	2.13 ± 0.88	2.73 ± 0.95	2.51 ± 1.08
*F*			316.372 ***	81.375 ***	6.067 ***
Post hoc			D < C < B < A	A < B < C < D	D < BC
PA on move	0 (A)	31.9 (16,447)	2.33 ± 0.94	2.62 ± 0.99	2.51 ± 1.12
	1–2 (B)	6.3 (3241)	2.28 ± 0.90	2.69 ± 0.96	2.53 ± 1.07
	3–4 (C)	7.4 (3802)	2.26 ± 0.91	2.67 ± 0.92	2.57 ± 1.06
	5–7 (D)	54.5 (28,146)	2.25 ± 0.90	2.69 ± 0.94	2.55 ± 1.08
*F*			27.575 ***	17.583 ***	6.319 ***
Post hoc			DCB < A	A < BD	A < CD

Notes: PA = physical activity; PE = physical education; * *p* < 0.05; *** *p* < 0.001.

**Table 5 behavsci-14-00160-t005:** Suicidal ideation, plan, attempt, and hopelessness experience according to types and levels of PA.

Type	Participants (n = 51,636)		Suicidal Ideation		Suicidal Plan		Suicidal Attempt		Hopelessness	
	Frequency (Days/Week)	% (n)	No % (n)	Yes % (n)	No % (n)	Yes % (n)	No % (n)	Yes % (n)	No % (n)	Yes % (n)
>60 min PA	0	33.1 (17,075)	28.4 (14,690)	4.6 (2385)	31.7 (16,355)	1.4 (720)	32.3 (16,674)	0.8 (401)	23.9 (12,347)	9.2 (4728)
	1–2	29.1 (15,028)	25.0 (12,898)	4.1 (2130)	27.9 (14,385)	1.2 (643)	28.3 (14,630)	0.8 (398)	20.7 (10,684)	8.4 (4344)
	3–4	21.1 (10,888)	18.0 (9305)	3.1 (1583)	20.1 (10,394)	1.0 (494)	20.5 (10,574)	0.6 (314)	14.9 (7690)	6.2 (3198)
	5–7	16.7 (8645)	14.4 (7446)	2.3 (1199)	15.9 (8201)	0.9 (444)	16.2 (8382)	0.5 (263)	11.7 (6035)	5.1 (2610)
χ^2^			2.378		12.961 **		13.366 **		20.227 ***	
Moderate PA	0	27.1 (13,972)	23.0 (11,868)	4.1 (2104)	25.8 (13,312)	1.3 (660)	26.2 (13,543)	0.8 (429)	19.0 (9830)	8.0 (4142)
	1–2	32.2 (16,652)	28.0 (14,438)	4.3 (2214)	31.0 (16,017)	1.2 (635)	31.5 (16,270)	0.7 (382)	23.1 (11,952)	9.1 (4700)
	3–4	22.8 (11,771)	19.6 (10,105)	3.2 (1666)	21.8 (11,235)	1.0 (536)	22.2 (11,484)	0.6 (287)	16.4 (8452)	6.4 (3319)
	5–7	17.9 (9241)	15.4 (7928)	2.5 (1313)	17.0 (8771)	0.9 (470)	17.4 (8963)	0.5 (278)	12.6 (6522)	5.3 (2719)
χ^2^			19.534 ***		27.383 ***		24.224 ***		11.382 *	
Vigorous PA	0	34.4 (17,739)	29.1 (15,016)	5.3 (2723)	32.7 (16,909)	1.6 (830)	33.4 (17,239)	1.0 (500)	24.3 (12,540)	10.1 (5199)
	1–2	28.0 (14,443)	24.3 (12,537)	3.7 (1906)	26.8 (13,858)	1.1 (585)	27.3 (14,104)	0.7 (339)	20.1 (10,358)	7.9 (4085)
	3–4	20.9 (10,798)	18.0 (9303)	2.9 (1495)	20.0 (10,346)	0.9 (452)	20.4 (10,516)	0.5 (282)	15.1 (7777)	5.9 (3021)
	5–7	16.8 (8656)	14.5 (7483)	2.3 (1173)	15.9 (8222)	0.8 (434)	16.3 (8401)	0.5 (255)	11.8 (3081)	5.0 (2575)
χ^2^			35.249 ***		15.828 **		9.992 *		11.460 **	
Strength training	0	54.5 (28,136)	46.4 (23,969)	8.1 (4167)	52.2 (26,932)	2.3 (1204)	53.1 (27,433)	1.4 (703)	38.9 (20,089)	15.6 (8047)
	1–2	21.1 (10,904)	18.3 (9443)	2.8 (1461)	20.2 (10,442)	0.9 (462)	20.5 (10,610)	0.6 (294)	15.0 (7762)	6.1 (3142)
	3–4	11.5 (5915)	9.9 (5106)	1.6 (809)	10.9 (5610)	0.6 (305)	11.1 (5728)	0.4 (187)	8.1 (4201)	3.3 (1714)
	5–7	12.9 (6681)	11.3 (5821)	1.7 (860)	12.3 (6351)	0.6 (330)	12.6 (6489)	0.4 (192)	9.1 (4704)	3.8 (1977)
χ^2^			25.241 ***		13.775 **		9.789 *		2.671	
Walks	0	4.2 (2161)	3.5 (1806)	0.7 (355)	3.9 (2014)	0.3 (147)	4.0 (2057)	0.2 (104)	3.0 (1571)	1.1 (590)
	1–2	7.0 (3594)	6.0 (3093)	1.0 (501)	6.7 (3434)	0.3 (160)	6.8 (3495)	0.2 (99)	5.0 (2601)	1.9 (993)
	3–4	10.2 (5282)	8.7 (4485)	1.5 (797)	9.7 (5023)	0.5 (259)	9.9 (5124)	0.3 (158)	7.2 (3693)	3.1 (1589)
	5–7	78.6 (40,599)	67.7 (34,955)	10.9 (5644)	75.3 (38,864)	3.4 (1735)	76.7 (39,584)	2.0 (1015)	56.0 (28,891)	22.7 (11,708)
χ^2^			15.253 **		33.604 ***		44.963 ***		9.026 *	
PE class PA	0	18.7 (9655)	15.7 (8116)	3.0 (1539)	17.7 (9142)	1.0 (513)	18.1 (9327)	0.6 (328)	12.9 (6665)	5.8 (2990)
	1	16.7 (8624)	14.5 (7501)	2.2 (1123)	16.0 (8267)	0.7 (357)	16.3 (8403)	0.4 (221)	11.9 (6152)	4.8 (2472)
	2	31.7 (16,349)	27.4 (14,132)	4.3 (2217)	30.3 (15,670)	1.3 (679)	30.9 (15,978)	0.7 (371)	22.6 (11,644)	9.1 (4705)
	Over 3	32.9 (17,008)	28.3 (14,590)	4.7 (2418)	31.5 (16,256)	1.5 (752)	32.1 (16,552)	0.9 (456)	23.8 (12,295)	9.1 (4713)
χ^2^			39.269 ***		22.264 ***		30.194 ***		32.047 ***	
PA on move	0	31.9 (16,447)	27.2 (14,068)	4.6 (2379)	30.3 (15,652)	1.5 (795)	30.9 (15,944)	1.0 (503)	22.2 (11,462)	9.7 (4985)
	1–2	6.3 (3241)	5.4 (2789)	0.9 (452)	6.0 (3083)	0.3 (158)	6.1 (3159)	0.2 (82)	4.5 (2314)	1.8 (927)
	3–	7.4 (3802)	6.3 (3249)	1.1 (553)	7.0 (3615)	0.4 (187)	7.2 (3697)	0.2 (105)	5.1 (2656)	2.2 (1146)
	5–7	54.5 (28,146)	46.9 (24,233)	7.6 (3913)	52.3 (26,985)	2.2 (1161)	53.2 (27,460)	1.3 (686)	39.4 (20,324)	15.1 (7822)
χ^2^			3.348		16.004 **		15.800 **		35.636 ***	
Total			85.9 (44,339)	14.1 (7297)	95.5 (49,335)	4.5 (2301)	97.3 (50,260)	2.7 (1376)	71.2 (36,756)	16.6 (14,880)

Notes: PA = physical activity; PE = physical education; * *p* < 0.05; ** *p* < 0.01; *** *p* < 0.001.

## Data Availability

No new data were created or analyzed in this study. Data sharing is not applicable to this study.

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
