# Peer review of "Associations between Physical Activity, Mental Health, and Suicidal Behavior in Korean Adolescents: Based on Data from 18th Korea Youth Risk Behavior Web-Based Survey (2022)"

_behavsci, 2024, doi:10.3390/bs14030160_

Round 1

Reviewer 1 Report

Comments and Suggestions for Authors

Dear Author

The manuscript deal an important aspect of currently discussed topic mental health and exercise among korean population. The reviewer has following comments to make about the manuscript:

INTRODUCTION

1. Though the introduction broadly states the aim of identifying physical activity status and its relationship with mental health and suicidal behaviours among Korean adolescents, it lacks specificity regarding the research questions or hypotheses, making it difficult for readers to grasp the study's precise focus.

2. In the introduction, adolescent experiences and mental health issues are encouraged to be discussed not in general terms but specifically addressing the unique socio-cultural factors affecting Korean adolescents, aside from academic stress.  

3. Include information as to how physical activity might influence mental health and reduce suicidal behaviours, leaving a gap in the conceptual framework.

4. Provide a review or critique of existing literature, particularly regarding the effectiveness of different types and levels of physical activity on mental health and suicidal behaviour among adolescents.

5. Avoid overfocusing on the negative outcomes associated with poor adaptation to adolescent challenges, but incorporate a more balanced view that considers protective factors and resilience, which could provide a more comprehensive understanding.

6. Key concepts such as "mental health problems," "suicidal behaviour," and "physical activity" are better to be provided with operational definitions for this study.  

7. Provide information about existing intervention strategies or how the study's findings might contribute to or differ from these strategies.

METHODS

1. The KYRBWS:  provide details about the validity and reliability of this specific survey instrument or its questions related to physical activity, mental health, and suicidal behaviour within the study population.

2. The author used the measures of mental health and suicidal behaviour to be broadly defined and rely on single-item questions, which may not fully capture the complexity of these constructs. The discussion on the selection of these specific measures and their limitations is absent.

3. Provide information about potential non-response bias or how the sample's characteristics might influence the generalizability of the findings to other adolescent populations outside South Korea.

4. Mention how potential confounders in the analysis were controlled during analysis, which is crucial for understanding the independent effects of physical activity on mental health and suicidal behaviour.

5. Provide information about what constitutes "moderate" versus "vigorous" physical activity or "self-rated health", which are not provided. This lack of clarity could affect the interpretation of the results and the replication of the study.

RESULTS

The author's presentation of the results is densely packed with statistical findings across multiple tables, making it challenging for readers to easily grasp the key findings. Simplifying the presentation or highlighting the most critical results could improve readability and comprehension.

DISCUSSION

1. The author has generalized the findings across genders, grades, and regions without adequately addressing the nuances or specific mechanisms that might explain these differences. It may also risk oversimplifying complex socio-cultural dynamics.

2. While comparing data from other countries, critically analyze why these differences might exist or how cultural, environmental, or policy factors in Korea specifically contribute to the observed patterns.

3. Provide some space for discussing theories related to adolescent development, stress, and coping, which could provide a deeper understanding of the observed relationships.

4. The author strongly emphasizes the role of physical activity in the discussion section without equally considering other important factors that affect adolescent mental health and suicidal behaviour, such as social support, academic pressure, and access to mental health services.

5. Discuss the cultural and social factors that may influence these differences, especially the societal expectations and pressures female adolescents face in Korea.

6. The discussion simplifies the complex nature of mental health and suicidal behaviour by focusing mainly on the frequency of physical activity. It does not adequately address the multifaceted nature of these issues or the need for comprehensive mental health interventions.

8. The discussion acknowledges limitations related to the analysis of types and frequencies of PA but does not address broader methodological limitations of the study, such as the use of self-reported data and the cross-sectional study design.

CONCLUSIONS

1. The author concludes that increasing PA levels will lower the risk of mental health issues and suicidal behaviour for all Korean adolescents, which might oversimplify the complex relationship between PA and mental health. It overlooks individual differences, such as the specific needs of adolescents with varying mental health conditions.

2. The author could not recommend specific, actionable recommendations on how to achieve this. Details on what these recommendations and policies might entail or how they could be implemented are not provided.

3. Suggest any strategy to address disparities in PA levels by gender and region.

4. The conclusion needs caution against the potential risks of excessive PA, such as physical injury or exacerbation of mental health issues due to overtraining.

5. There is an absence of a call for further research.

Author Response

Dear reviewer:

Reviewer 2 Report

Comments and Suggestions for Authors

General comments

This study aims to identify the current status and relationship between physical activity (PA), mental health, and suicidal behavior among Korean adolescents and recommend appropriate PA types and levels to lower the risk of mental health and suicidal behavior among adolescents. The manuscript is well written and presents some novel information that has the merit to be disseminated within the health field.

The state of the art needs to be improved, expanding the bibliographic support.

In the following I will comment specifically on the improvements to be made.

Specific comments

Introduction

Physical activity is also related to sleep and with the socioeconomic index (https://doi.org/10.3390/children10030551). Sleep efficiency and socioeconomic index is one of the variables that will be implicitly related to mental health and suicidal behavior. Contributions in this regard should be made in the state of the art. In the case of having these variables in the survey, I would suggest including it in the analyses. Cites 42 and 44 refer to this information.

I am attaching 4 references to make the authors aware of the influence of physical activity and sleep, together with the risk of stress and depression, parameters closely related to mental health and suicidal risk, the main topic of the article:

  Yeo, S.C.; Tan, J.; Lo, J.C.; Chee, M.W.L.; Gooley, J.J. Associations of time spent on homework or studying with nocturnal sleep behavior and depression symptoms in adolescents from Singapore. Sleep Health 20206, 758–766.
  Ghrouz, A.K.; Noohu, M.M.; Manzar, D.; Spence, D.W.; BaHammam, A.S.; Pandi-Perumal, S.R. Physical activity and sleep quality in relation to mental health among college students. Sleep Breath. 201923, 627–634.   Atoui, S.; Chevance, G.; Romain, A.J.; Kingsbury, C.; Lachance, J.P.; Bernard, P. Daily associations between sleep and physical activity: A systematic review and meta-analysis. Sleep Med. Rev. 202157, 101426.
  Park, S. Associations of physical activity with sleep satisfaction, perceived stress, and problematic Internet use in Korean adolescents. BMC Public Health 201414, 1143.

The introduction is short; both the content and the bibliographic support of some statements should be expanded.

Material and methods

The instrument for measuring the variables should be added, even if only as supplementary material. The wording of the possible responses of the variables becomes difficult to read, I would suggest adding it in table-figure format.

Modify line 129 by: The significance level for all statistical analyses was established at 0.05 (p≤0.05).

The procedure should be better explained.

The number of the ethics committee with which the data were recruited should be named and added in the sample section.

Results

The results are well analyzed

In all .05 - .01 or .001 a 0 must be added in front.

Check the numbers in all tables, that they all contain 2 decimal places, there are some numbers with one decimal place, others with two, others with 3...

Discussion and conclusions

The discussion is very good, congratulations.

I would add a paragraph with future lines of research.

What practical applications do you see for your research? Add them.

 All the best in your submission!

Author Response

Dear reviewer:

Reviewer 3 Report

Comments and Suggestions for Authors

Dear Authors of "Associations between Physical Activity, Mental Health, and Suicidal Behavior in Korean Adolescents: Based on Data from the 18th Korean Youth Risk Behavior Web-based Survey (2022)",

Upon thorough review of your manuscript, I extend my congratulations for a significant study on a critical topic. Your focus on the interplay between physical activity, mental health, and suicidal behavior in adolescents is both relevant and timely.

However, I believe there are areas for enhancement to improve comprehension and impact:

-        Clarity in Tables: More detailed and clearer presentation of tables would greatly aid readers in understanding your results and facilitate deeper analysis.

-        Inclusion of Graphs: Incorporating descriptive graphs would be immensely beneficial for visualizing identified relationships and trends. Graphs can powerfully highlight key findings and enhance the overall comprehensibility of your article.

-        Methodological Details: While your study provides a good overview of the methods used, a deeper dive into specific methodological aspects could strengthen confidence in your results and analysis.

-        Expanded Discussion: Enriching the discussion section to more deeply connect your findings with existing literature and potential practical applications would be valuable. This could include a more detailed consideration of the implications of your findings for policies or interventions aimed at adolescents.

In conclusion, your article is a significant contribution to the field, and with these suggested improvements, I believe your work can achieve greater impact and recognition in the scientific community. I recommend a minor revision to incorporate these suggestions.

Thank you for the opportunity to review your work, and I look forward to seeing your completed study published.

Sincerely,

Author Response

Dear reviewer:

Round 2

Reviewer 1 Report

Comments and Suggestions for Authors

The author addressed the comment adequately, and it may be considered for acceptance.

Reviewer 2 Report

Comments and Suggestions for Authors

Thank you for taking my contributions into consideration.